# Live Traps for Adult Brown Marmorated Stink Bugs

**DOI:** 10.3390/insects10110376

**Published:** 2019-10-29

**Authors:** David Maxwell Suckling, Mary Claire Levy, Gerardo Roselli, Valerio Mazzoni, Claudio Ioriatti, Marco Deromedi, Massimo Cristofaro, Gianfranco Anfora

**Affiliations:** 1Technology Transfer Center, Fondazione Edmund Mach, I-38010 San Michele all’Adige (TN), Italy; gerardoroselli@hotmail.it (G.R.); claudio.ioriatti@fmach.it (C.I.); 2The New Zealand Institute for Plant and Food Research Ltd., PB 4704 Christchurch, New Zealand; 3School of Biological Sciences, University of Auckland, 1072 Auckland, New Zealand; 4Kallisto, Mt Pleasant, 8081 Christchurch, New Zealand; clairelevy@xtra.co.nz; 5Biotechnology and Biological Control Agency, 00123 Rome, Italy; m.cristofaro55@gmail.com; 6Center of Agriculture, Food and Environment (C3A), University of Trento, I-38010 San Michele all’Adige (TN), Italy; gianfranco.anfora@fmach.it; 7Research and Innovation Center, Fondazione Edmund Mach, I-38010 San Michele all’Adige (TN), Italy; valerio.mazzoni@fmach.it (V.M.); marco.deromedi@fmach.it (M.D.); 8National Agency for New Technologies, Energy and Sustainable Development (ENEA), 00123 Rome, Italy

**Keywords:** aggregation pheromone, *Halyomorpha halys*, trap, lure and kill, sterile insect technique, wild harvest

## Abstract

Surveillance for detection of the brown marmorated stink bug, *Halyomorpha halys*, is reliant on sticky panels with aggregation pheromone, which are low cost, but very inefficient (est. 3%). Trapping for adults was conducted in Italy with novel live (or lethal) traps consisting of aggregation pheromone-baited cylinders with a wind vane, with the upwind end covered by mesh and the downwind end sealed by a removable entry-only mesh cone, admitting the attracted bugs. The novel traps caught up to 15-times more adult *H. halys* than identically-baited sticky panels in two weeks of daily checking (n = 6 replicates) (the new live traps were, in Run 1, 5-, 9-, 15-, 13-, 4-, 12-, 2-fold; and in Run 2, 7-, 1-, 3-, 7-, 6-, 6-, and 5-fold better than sticky traps, daily). The maximum catch of the new traps was 96 live adults in one trap in 24 h and the average improvement was ~7-fold compared with sticky panels. The rotating live traps, which exploit a mesh funnel facing the plume downwind that proved useful for collecting adults, could also be used to kill bugs. We expect that commercially-available traps could replace the crude prototypes we constructed quickly from local materials, at low cost, as long as the principles of a suitable plume structure were observed, as we discuss. The traps could be useful for the sterile insect technique, supporting rearing colonies, or to kill bugs.

## 1. Introduction

*Halyomorpha halys* (Stål, 1855), the brown marmorated stink bug (BMSB), is a highly invasive species from Southeast Asia, which feeds on field crops, vegetables, tree fruits, and nuts and ornamentals. Both nymphs and adults feed on developing and ripe fruits and seeds and can cause severe damage to crops [1,2]. In climate conditions such as Northern Italy or much of USA [3], BMSB is predicated to have two or more generations per year [4]. Pre-reproductive adults over-winter inside houses and release an unpleasant odor when disturbed [5]. The BMSB is increasingly widespread in Europe [1,6]. In Northern Italy alone, the local associations of fruit production have estimated about €150 million/year losses in 2016–2018. Interceptions of this species have been reported in other countries [7] including New Zealand [8]. No successful eradication attempts of established populations have been reported thus far.

A two-part aggregation pheromone ((3S,6S,7R,10S)-10,11-epoxy-1-bisabolen-3-ol and (3R,6S,7R,10S)-10,11-epoxy-1-bisabolen-3-ol) [9] has been synergized with the addition of methyl (*E*,*E*,*Z*)-2,4,6-decatrienoate (MDT) [10]. Various trapping systems have been investigated for integrated pest management (IPM) [11] and, based on these lure surveillance systems, have been tested for border protection, because of high interception rates in New Zealand [8]. For jurisdictions such as New Zealand that conduct surveillance, new and effective methods for both surveillance and suppression are important. Sticky panels are low cost, but could suffer from poor bug retention and loss of trap efficiency owing to the accumulation of dust, leaves and detritus. Other trap types have been investigated [12], although clear sticky panels were recently reported as suitable for BMSB monitoring and detection, by a New Zealand government-supported study in the USA [13]. Most of the traps reported so far appear generally similar in functional concept as variations on a theme, and produce fairly similar results [14]. Simply hanging pyramid traps in trees did not result in an improvement, although placing the lures in such a way to increase airflow (outside the trap) did improve results about two-fold [12], which is fairly unsurprising. None of the treatments were particularly successful improvements.

Live traps for harvesting wild insects can support research in various ways, by offering efficient field collection [15]. A cylindrical trap design originally designed for live trapping tortricid moths in a mark-release recapture study [16] was physically enlarged to take account of observations on BMSB behavior in Italy, and a prototype was built and tested. This proved successful (with a catch of 50 adult BMSB in the first 24 h), so a replicated trial was established to compare these traps with the proposed surveillance system for New Zealand, and this communication was produced to widen the debate about surveillance efficiency as well as suppression of this unwanted organism.

## 2. Materials and Methods

### 2.1. Comparing the Live Traps and Sticky Panels

The aim of the research was to compare the catch efficiency of the highly mobile adult BMSB population from baited sticky panels and novel BMSB live traps, in the limited time available. Trapping studies for adult (and potentially nymph stages) of BMSB were conducted in late August 2019 at Fondazione Edmund Mach (46°11′43″ N, 11°8′5″ E), San Michele all’Adige, Trentino, Italy. We evaluated captures with two different kinds of traps, Pherocon sticky panels (Trécé, Adair, OK, USA), with high dose lures (Trécé, Adair, OK, USA), which contained 200 mg (i.e., 4-fold loading) of two component aggregation pheromones, (3S,6S,7R,10S)-10,11-epoxy-1-bisabolen-3-ol and (3R,6S,7R,10S)-10,11-epoxy-1-bisabolen-3-ol [9], plus methyl (*E*,*E*,*Z*)-2,4,6-decatrienoate [10]. The lures were positioned on top of the sticky panels, on overhanging branches. This combination of lure and sticky panel trap has been under investigation for surveillance in New Zealand because of a high rate of interceptions [8]. The second trap type [16] rotated using a wind vane, while trap cylinders were constructed from soft plastic plant pots (30 cm diameter × 40 cm high) with the bottoms cut off and the tops joined and sealed. The upwind end was covered with a flat panel of stainless gauze (1 mm mesh size) and sealed using hot glue, with the lure hung ~5 cm inside the opening of the center of the trap. The downwind end was sealed with a removable stainless mesh cone. The wind vane was added to the top of the downwind end as a vertical pane constructed from corflute (corrugated plastic), and the trap was suspended and able to pivot balanced on a string suspended from a tree branch (Figure 1).

We set the traps (n = 6 replicates, alternating trap types at 10 m spacings) at a sloping forest margin with adjacent vineyards of mixed grape varieties downhill. The experiment was set up on 22 August 2019 and checked daily for seven days (one run), before moving to a similar nearby location, 200 m along and 50 m higher, for a second identical run (traps were re-randomized). Catches in traps were sexed, counted, and removed daily. The large number of catches of live BMSB in the prototype live traps (up to 96 adult bugs per 24 h) required the trap contents to be emptied into a large plastic bag (80 cm × 1 m), and bugs were then individually sexed and removed around 9 am daily onto a second bag for transport to the laboratory culture. The smooth internal surface of the traps expedited emptying them. A short YouTube video accompanies this article to illustrate the new trap (Appendix A).

### 2.2. Statistical Methods

On the basis of experience with two key factors affecting insect trapping, we balanced our design with replicates and days about evenly. Summed catches for each trap were log-transformed to generate an approximately normal distribution, in order to stabilize the variance (*p* = 0.198 for normality after transformation) [17]. Catches were then compared for significant differences by type and replicate using a general linear model in Minitab (v18) [17].

## 3. Results

Both types of traps were successful at catching BMSB adults (Figure 2, Appendix A), and the catches initially increased daily after installation of the live traps and, to a much lesser extent, in sticky traps, but varied day to day without obvious weather influence (see weather data, Appendix A). Few nymphs were caught with either trap, so the results presented have been confined to adults for brevity. The analysis of variance (ANOVA) was highly significant for trap type, with a lower level of significance for replicate. Numbers caught reduced on days five to seven during Run 1. Numbers were lower at the location for the second run, located at about 50 m higher elevation and 200 m away, with a total of 1061 caught in Run 1 and 555 in Run 2.

The difference in trap efficiency among traps varied daily, but the live trap was better than sticky panel traps on 13/14 of the days. Low catches or zero catches occurred with both trap types (Figure 3), and were frequent with the sticky traps, while the live traps caught up to 96 adults per day, with nine trap counts over 30 BMSB per day. The sex ratio was similar between trap types, and significantly biased (1.6 and 2.2 females per male for the live traps (X^2^ = 73.5, *p* < 0.0001) and sticky panels (X^2^ = 55.6, *p* < 0.0001), respectively).

Catch varied by transect position, with the lowest catches in internal traps (Figure 4), suggesting trap competition. Trap position appeared to have little effect on the sticky panels.

## 4. Discussion

Effective pest monitoring is an essential tool for IPM, and pheromones have provided a tremendous boost to this field by attracting insects directly to traps. Efficient traps can support management decisions, to restrict the use of insecticides and reduce costs, non-target effects, and secondary pest outbreaks. The need for understanding the meaning of a stink bug in a trap in IPM generally aligns with the biosecurity detection needs in countries like New Zealand, where high sensitivity at first detection will be essential for delimitation in a response [8]. The highly successful lures for BMSB have opened new opportunities for IPM as well as surveillance, but traps require more than simply attraction to be effective. Trap efficiency should remain relatively constant over time and, unfortunately, traps based on sticky panels are likely to eventually suffer from loss of efficacy through the accumulation of dust, leaves and detritus. Clear sticky panels are reported as suitable for BMSB monitoring and detection [13], highlighting a lack of better alternatives.

At the forest–vineyard margin, the numbers of bugs sampled (alive and potentially killed or added to the laboratory culture) were substantially higher for the live traps than for the sticky traps. Neither trap caught many nymphs, which was in part because of the tendency of walking nymphs to avoid the sticky glue, although nymphs were observed to walk down the string to the live traps, and numbers on the outside of the traps apparently increased over time.

Catches of BMSB adults (and nymphs) typically build up over the first three days with the aggregation pheromone, independent of the trap type (Suckling et al., unpublished results). Catches in other trials (Suckling et al., unpublished data) have remained relatively steady thereafter, but here, a mid-week peak was twice followed by a >50% decline for the last three days, suggesting that an influence of adult removal could be having an effect on the local population, but this would need to be verified with absolute sampling techniques.

Our very short trial was resource-limited, but the results are sufficient to encourage further examination of this concept of live trapping, which avoids the need for a beating tray and other physical collection methods. This method of live collection has already added hundreds of insects to the colony at Fondazione Edmund Mach, and plans have been developed to test the approach further using commercially-available materials. There was a sex bias towards female BMSB in both traps, which is a useful finding because it is unlikely that both traps are biased and fail to represent the actual sex ratio. The live traps showed evidence of trap competition, as higher catches were generally made towards the ends of the transects, which is equivalent to the corner trap in an array, which faces less competition than central traps [18]. The sticky panels did not show this effect, suggesting that the lower catch efficiency could be masking the effect. While U.S. researchers have preferred to use 50 m trap spacings, their objectives were different, such as seeking calibration in absolute values of trap efficiency. Here, we sought to compare two trap types and to demonstrate the potential of live capture, rather than pursue the theoretical active space [19]. Leskey et al. [20] and Rice et al. [12] categorically state that “A killing agent is necessary for successful trapping of *H. halys* using pyramid traps with collection jars”, but perhaps the current work creates new options through design.

Live catches in this type of trap could be used for collection of insects aggregating pre-winter, as well as for irradiation and release of males for the sterile insect technique (SIT) [21], and it may even be possible to generate overwintering behavior if suitable substrates are provided within the traps in late autumn. Males in particular are needed for SIT, so physical sorting by sex would still be necessary.

Such traps could also readily be converted into killing stations through the addition of long-life insecticide netting into the body of the trap, especially at the upwind panel where we observed the highest numbers of adults caught inside the traps (Figure 1). This type of technology could prove useful in an eradication of a delimited population. Sticky panels are currently preferred for surveillance in New Zealand [8], but proactively established killing stations could complement this in high-risk sites. While our study of the new traps did not directly investigate the full potential for killing BMSB adults, we can conclude that this trap concept could be compatible with the needs of an eradication to reduce the numbers of a known and delimited population, while avoiding broadcast insecticide use. This type of mass trapping approach has the benefits of limiting broadcast insecticide usage and compatibility with biological control, but has material costs and ongoing labor costs [22], although these costs could be sustained for the duration of an eradication [23]. Identification of a suitable commercially-available low-cost fish trap or similar trap is proposed, now that the principle of this trap type has been demonstrated. Solid cylindrical walls, a mesh funnel opening downwind with a small non-return opening, a wind vane, and pivotal suspension are expected to be critical design features.

Reported observations of BMSB behavior around traps have focused on a range of attributes including walking [11], but have apparently failed to consider the possibility that the plume can (hypothetically) be delivered from inside a cylinder, and apparently reach the population more successfully by extending the plume further than with lures that are fully exposed in the open air, for example, around a sticky trap. The mechanistic role of plume structure warrants further investigation, as the principle is general and may well enable improvement in catches to other lures and insects. Once in vogue for moth pheromones [24,25,26], the literature on plume structure and insect behavior in and around pheromone-baited traps has reduced in recent decades, but perhaps the field needs review for other orders, as previous extensive research on BMSB trapping has apparently failed to discover how to greatly improve catches, as reported here.

The new trap concept may have considerable relevance to IPM, where its potential to help growers can now be evaluated within invaded jurisdictions. Recent approaches such as aggregating insects to sacrificial killing trees [27] could be potentially be replaced with more effective “lure and kill” traps to reduce environmental exposure from insecticides, if the initial promise of our new traps can be confirmed. An average improvement of ~7-fold and up to 15-fold over the recommended sticky panels appears to warrant further investigation for detection and surveillance, as well as for IPM, but needs to be placed in the context of overall insect population suppression.

## 5. Conclusions

The new traps were surprisingly effective at live trapping BMSB given extensive research on trapping this insect since the aggregation pheromone was identified. The new trap concept could represent a useful tool with a range of applications, although as yet there is limited understanding of why the trap design is working so well. More work on aspects of the trapping system (size, design, plume structure) would be useful to extract the essentials for development of even more effective traps in future. The immediately obvious applications of live capture include wild harvest for supporting rearing colonies, as well as supporting our objective of the sterile insect technique, because mass rearing is a currently a limitation to that approach. Equally, the traps could provide the basis for a more sensitive surveillance system in biosecurity, given the much-improved catch rates.

## Figures and Tables

**Figure 1 insects-10-00376-f001:**
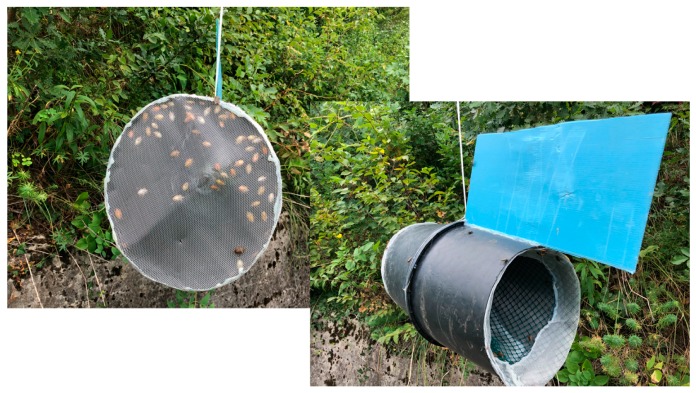
Rotating live trap for *Halyomorpha halys* consisting of two large black flower pots joined in the middle, with bases removed and a removable mesh cone for bug entry and removal, an upwind mesh panel for airflow, and an aggregation pheromone lure, with a wind vane to generate the best plume from the trap.

**Figure 2 insects-10-00376-f002:**
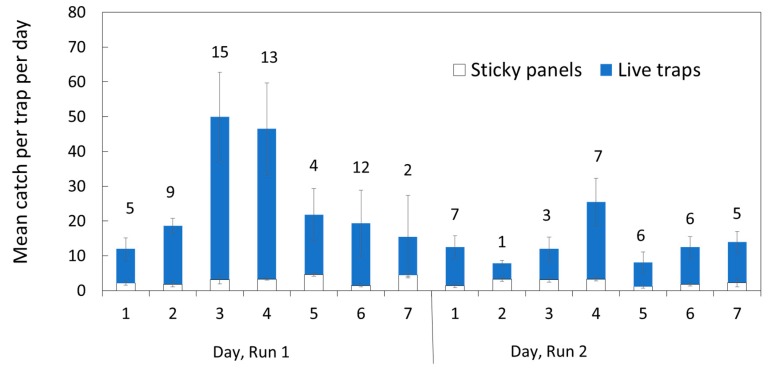
Mean daily catch per trap of adult *Halyomorpha halys* in alternating sticky panels and live traps, on a vineyard–forest margin at Fondazione Edmund Mach, San Michele all’Adige (TN), Italy. Error bars show one standard error (n = 6 replicates). Runs 1 and 2 were at different locations, 200 m apart. Labels indicate the mean improvement in catch from the live traps over sticky panels.

**Figure 3 insects-10-00376-f003:**
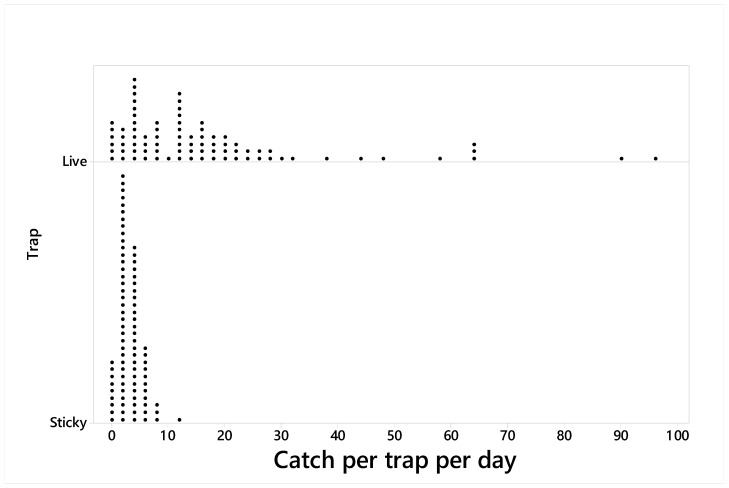
Dotplot of catch per trap per day with spatially-alternating live traps (live, upper) and sticky panels (lower) at the forest–vineyard margin in San Michele all’Adige (TN), Italy (n = 6 reps, operated daily for two runs of seven days, at 200 m spacings between runs).

**Figure 4 insects-10-00376-f004:**
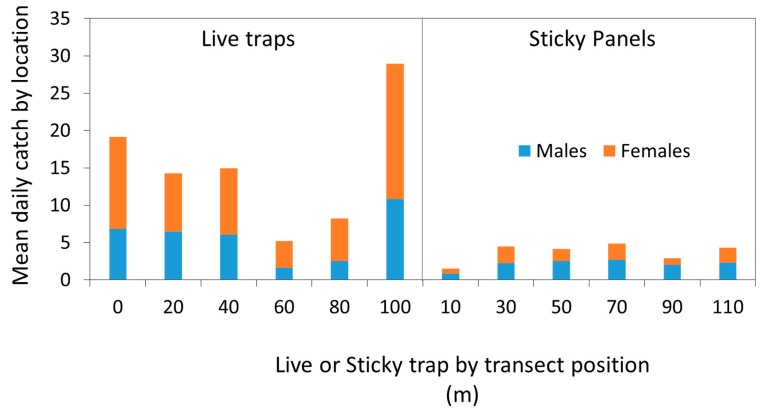
Effect of transect positions on mean daily catch of *Halyomorpha halys* by sex and trap type at the forest–vineyard margin in San Michele all’Adige (TN), Italy (n = 6 replicates, operated for 14 days (runs combined)).

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
