# Peer review of "Live Traps for Adult Brown Marmorated Stink Bugs"

_insects, 2019, doi:10.3390/insects10110376_

Round 1
Reviewer 1 Report
In this manuscript the authors evaluated a novel trap for the monitoring of the brown marmorated stink bug, Halyomorpha halys. This is an invasive species of global impact on agricultural economy. This considering, the manuscript might be of high interest for people and researchers involved in pest monitoring and control. Overall, the manuscript is well written (but there are few typos) and the methods for data collection have no remarkable flaws. However, the quality of the figures should be improved. I suggest adding a verb in the title, for example: “Highly Efficient Live Traps to monitor Adult Brown Marmorated Stink Bugs”. Below some specific comments:
L39: Capitalize the word “northern”
L115-116: please, include statistical analysis (e.g. Chi-squared or Fisher’s test)
Figure 2: I would correct the y-axis label with “Catches per trap per day”. In addition, I suggest the use of two different colours for the bars and the removal of the long list of “Sticky panels – Live traps” series
Figure 4 caption: please, add information about “S” and “L” used in the x-axis labels. In addition, a comparison of males vs. females catches between the two trap types should be included. The information can be reported in L 115-117.
Figure 5: please, increase the thick of the x-axis line
L160: what does the word “reference” mean here?
L162: please, correct the typo in “techniques”
Author Response
Reviewer 1
In this manuscript the authors evaluated a novel trap for the monitoring of the brown marmorated stink bug, Halyomorpha halys. This is an invasive species of global impact on agricultural economy. This considering, the manuscript might be of high interest for people and researchers involved in pest monitoring and control. Overall, the manuscript is well written (but there are few typos) and the methods for data collection have no remarkable flaws. However, the quality of the figures should be improved. I suggest adding a verb in the title, for example: “Highly Efficient Live Traps to monitor Adult Brown Marmorated Stink Bugs”.
I disagree with this suggested title change as it limits the utility - where actually mass live collection for SIT is part of the plan for these traps, this is not monitoring.
Below some specific comments:
L39: Capitalize the word “northern”
Ok
L115-116: please, include statistical analysis (e.g. Chi-squared or Fisher’s test)
Now added
Figure 2: I would correct the y-axis label with “Catches per trap per day”.
The term Catch per trap per day is preferred English and more standard in my view.
In addition, I suggest the use of two different colours for the bars and the removal of the long list of “Sticky panels – Live traps” series
Revised
Figure 4 caption: please, add information about “S” and “L” used in the x-axis labels. In addition, a comparison of males vs. females catches between the two trap types should be included. The information can be reported in L 115-117.
Revised
Figure 5: please, increase the thick of the x-axis line
Removed
L160: what does the word “reference” mean here?
Reworded
L162: please, correct the typo in “techniques”
OK
Reviewer 2 Report
Review of Suckling et al. Short communication for novel trap design
Design of a new trap for BMSB is quite exciting, and the authors have performed a quick evaluation of a novel and inexpensive trap for an important pest. The incorporation of a windvane to help disperse the pheromone plume is quite interesting and the novel trap design, while mimicking many of the original bucket collection containers (see Cotrell) appears to hold some promise. However, the authors have critical experimental errors that invalidate their findings. Most importantly is that while testing the aggregation pheromone and synergist for BMSB, the authors have placed traps at 10m spacing instead of the standard 50m spacing. Multiple refereed papers use the 50m spacing between traps to ensure independent results, especially when comparing trap designs. This spacing has further been supported by Kirkpatrick et al. 2019 (Environmental Entomology doi: 10.1093/ee/nvz093) that identified the plume reach of the pheromone to be <3m but the maximum dispersive distance to be 40-70m which should strongly influence trap placement. Placing traps closer can lead to interference of the pheromone plume as perceived by the insect, making each sampling unit (trap treatment) not independent of the other trap treatment. There is also insufficient replication of the trap types.
As a season-long pest, and with nymphs being quite important from a monitoring and management perspective, investigating a novel trap design for a short 2-week period that does not include a period in which nymphs are abundant, does not provide sufficient data. Ensuring that a trap mimics the insect’s phenology is critical to surveillance and IPM decisions, as the authors reference in the Discussion. Previous work on the sticky panel trap indicates that it does reflect phenology, despite lower trap catch (see Acebes Doria et al. 2019 Econ Entomol which compares pyramid and sticky panels for phenology at multiple locations in the US). This study does not accomplish that with a 2-week study. I see this study as very preliminary and in need of additional season-long data with sufficient replication and trap spacing, even for a Short Communication.
Specific comments:
Title: replace “Highly Efficient” with “Alternative”. There are multiple times in the manuscript where the authors use highly persuasive or inflammatory terms that detract from the science.
Abstract:
Ln 19: define “inefficient” in this context, authors could instead use data, that the sticky panels trap about 3% of the surrounding population.
Introduction:
Ln 40: BMSB has two generations throughout the US as well and should be reported as such. See Nielsen et al. 2016; 2017 Frontiers in Physiology doi.org/10.3389/fphys.2016.00165; doi.org/10.3389/fphys.2017.00568
Ln 48: Use of the two-part pheromone plus MDT has “synergized” attraction and trapping efforts, not improved
Ln 54-55: Add citation for poor retention of sticky panel
Ln 58: remove “apparently”
Methods:
The spacing between traps, especially when using the 4x high rate lure, is too small to have independence between sampling units. Because of this, the authors cannot be certain that differences observed between traps are due to the trap design, insect behavior or random chance. Also, there are only 2 replicates. In trapping studies, especially when using an aggregation pheromone, a transect of traps (ie. 6) is used and then replicated in space and/or time. Were traps rotated weekly to avoid effects of trap location since BMSB has an aggregation dispersion in the field?
Ln 78: How was the sticky panel deployed? Acebes-Doria 2018 (Insects doi:10.3390/insects9030082) compared pyramid vs. sticky on stakes. Rice et al (Env. Entomol doi: 10.1093/jee/toy185), compared multiple trap designs including a ground deployed pyramid and a hanging pyramid and found that ground deployed had significantly higher catch
Results:
I find many of the figures to be duplicative of each other. For example, Fig 2 and 4 tell the same information, with the addition of sex. Figure 5 is too busy to interpret and should be removed. The data should be shown as Means (between the 6 traps and locations), by day with standard error.
Table S1 is never referenced
Ln 119: Change to Supp Table 2
Discussion:
Ln 148: Traps (in this case) are not attractive, you are comparing the capture between trap types
Ln 160: a citation is needed to support this statement. Without a statistical test, which is needed, the numbers pre- and post-peak (day 3) look very similar to me. Again, there in insufficient replication and serious flaws in the trap deployment that do not allow for interpretation of the data
Ln 168-172: BMSB stops responding to the aggregation pheromone when seeking out overwintering sites. I believe the authors are referring to the high numbers observed in traps before thus time period and the text should be adjusted appropriately.
Ln 192-197: See Kirpatrick et al 2019 (referenced above) on plum reach of the BMSB pheromone
Author Response
Review 2
Design of a new trap for BMSB is quite exciting, and the authors have performed a quick evaluation of a novel and inexpensive trap for an important pest. The incorporation of a windvane to help disperse the pheromone plume is quite interesting and the novel trap design, while mimicking many of the original bucket collection containers (see Cotrell) appears to hold some promise. However, the authors have critical experimental errors that invalidate their findings. Most importantly is that while testing the aggregation pheromone and synergist for BMSB, the authors have placed traps at 10m spacing instead of the standard 50m spacing. Multiple refereed papers use the 50m spacing between traps to ensure independent results, especially when comparing trap designs. This spacing has further been supported by Kirkpatrick et al. 2019 (Environmental Entomology doi: 10.1093/ee/nvz093) that identified the plume reach of the pheromone to be <3m but the maximum dispersive distance to be 40-70m which should strongly influence trap placement. Placing traps closer can lead to interference of the pheromone plume as perceived by the insect, making each sampling unit (trap treatment) not independent of the other trap treatment.
I have published a lot on transects and also grids with models to determine overlapping trap competition effects and note that different projects require different perspectives.
These concerns imply a misunderstanding of the aims of the work, which are different from the desire of US IPM specialists to have absolute calibration of trapping efficiency from any trap in use, which has expended huge resources in their system.
In the contexts of biosecurity surveillance or mass field collection, the rules are different and absolute trap efficiency is only of value to determine trap spacing in a grid, no matter which trap is chosen. The grid spacing would indeed be driven by the plume research (which partly funded it from NZ), but the uses proposed for these new live traps are different because a relative index of improvement is sufficient here.
There is also insufficient replication of the trap types.
It is not clear what mathematical basis there is for this, the replication level is included in the statistical analysis that determines the presence of significant differences. On 13/14 occasions an improvement was evident, averaging seven fold.
As a season-long pest, and with nymphs being quite important from a monitoring and management perspective, investigating a novel trap design for a short 2-week period that does not include a period in which nymphs are abundant, does not provide sufficient data.
It depends on your purpose. Actually I have a whole other study on a new nymph sampling trap that is different again from the sticky traps used routinely. Some nymphs are caught on sticky panels on overhanging tree branches and a number were caught on the outside of the live traps. A purpose-built trap for killing nymphs will be reported next.
Ensuring that a trap mimics the insect’s phenology is critical to surveillance and IPM decisions, as the authors reference in the Discussion. Previous work on the sticky panel trap indicates that it does reflect phenology, despite lower trap catch (see Acebes Doria et al. 2019 Econ Entomol which compares pyramid and sticky panels for phenology at multiple locations in the US). This study does not accomplish that with a 2-week study.
Any trap which catches seven times more bugs will give better phenology if serviced diligently. Another data set shows phenology with nymphs using another new trap design, in a similar way to this study which focussed on adults, partly for autumn mass collection (a completely different agenda from the perspective of the US reviewer).
Sampling nymphs for surveillance using sticky panels proved very inefficient. This comments is defending a very inefficient system but there are alternatives. I could only sample what the resources allowed, and the progress made is actually impressive compared with the resources expended in USA. This will become more obvious with two additional papers on BMSB intended for this journal.
I see this study as very preliminary and in need of additional season-long data with sufficient replication and trap spacing, even for a Short Communication.
I note that two reviewers advocated publication. There are enough data here to defend and to raise international interest, given an average of seven-fold improvement over the established “best” technology.
I see a need for new ideas to be presented quickly because of the cost of the pest to farmers in many countries, present and future jurisdictions included.
Specific comments:
Title: replace “Highly Efficient” with “Alternative”. There are multiple times in the manuscript where the authors use highly persuasive or inflammatory terms that detract from the science.
Title changed to Live Traps for Adult Brown Marmorated Stink Bugs
Abstract:
Ln 19: define “inefficient” in this context, authors could instead use data, that the sticky panels trap about 3% of the surrounding population.
OK
which are low cost but very inefficient (est. 3%).
Introduction:
Ln 40: BMSB has two generations throughout the US as well and should be reported as such. See Nielsen et al. 2016; 2017 Frontiers in Physiology doi.org/10.3389/fphys.2016.00165; doi.org/10.3389/fphys.2017.00568
Ln 48: Use of the two-part pheromone plus MDT has “synergized” attraction and trapping efforts, not improved
changed
Ln 54-55: Add citation for poor retention of sticky panel
could suffer from poor bug retention and loss of trap efficiency due to the accumulation of dust and detritus
Ln 58: remove “apparently”
Done
Methods:
The spacing between traps, especially when using the 4x high rate lure, is too small to have independence between sampling units. Because of this, the authors cannot be certain that differences observed between traps are due to the trap design, insect behavior or random chance.
This is incorrect. Even if the traps compete with each other due to overlapping active space, this does not reduce the value of the comparison because the aims are different and the spacing was consistent. Differences in traps leading to a consistent seven-fold improvement over sticky panels are unlike to be due to random chance.
Also, there are only 2 replicates.
This is incorrect (the transect is not the scale of replication, it is the trap)
In trapping studies, especially when using an aggregation pheromone, a transect of traps (ie. 6) is used and then replicated in space and/or time. Were traps rotated weekly to avoid effects of trap location since BMSB has an aggregation dispersion in the field?
Traps were re-randomised.
Ln 78: How was the sticky panel deployed? Acebes-Doria 2018 (Insects doi:10.3390/insects9030082) compared pyramid vs. sticky on stakes. Rice et al (Env. Entomol doi: 10.1093/jee/toy185), compared multiple trap designs including a ground deployed pyramid and a hanging pyramid and found that ground deployed had significantly higher catch
Results:
I find many of the figures to be duplicative of each other. For example, Fig 2 and 4 tell the same information, with the addition of sex.
Figures revised and improved.
On closer inspection is it not the case that one is over time and one is over space?
Figure 5 is too busy to interpret and should be removed. The data should be shown as Means (between the 6 traps and locations), by day with standard error.
Fig 5 has been deleted.
Table S1 is never referenced
Done
Ln 119: Change to Supp Table 2
Discussion:
Ln 148: Traps (in this case) are not attractive, you are comparing the capture between trap types
The visual component has not been investigated
Ln 160: a citation is needed to support this statement. Without a statistical test, which is needed, the numbers pre- and post-peak (day 3) look very similar to me. Again, there in insufficient replication and serious flaws in the trap deployment that do not allow for interpretation of the data
Ln 168-172: BMSB stops responding to the aggregation pheromone when seeking out overwintering sites. I believe the authors are referring to the high numbers observed in traps before thus time period and the text should be adjusted appropriately.
Ln 192-197: See Kirpatrick et al 2019 (referenced above) on plum reach of the BMSB pheromone
Reviewer 3 Report
The research provides some interesting information on a new type of trap that can help in the fight against a particularly harmful insect. I admit that found some difficulty in reading and understand the text, not for the English form, but for the way in which the sentences were constructed. In some cases they are too long and it is difficult to follow them. For the rest it seems to me an interesting research that surely should be encouraged to further development to perfect the method.

Author Response
Review 3
The research provides some interesting information on a new type of trap that can help in the fight against a particularly harmful insect. I admit that found some difficulty in reading and understand the text, not for the English form, but for the way in which the sentences were constructed. In some cases they are too long and it is difficult to follow them. For the rest it seems to me an interesting research that surely should be encouraged to further development to perfect the method.
Improvements as above.
Round 2
Reviewer 2 Report
The authors have developed a novel trap for H. halys which they report catches 7-fold higher number of adults. Although a significant amount of work has occurred on trap design and lure development, there is still a lot of room for growth in this context. The authors trap design looks quite promising for increasing numbers in traps and is an exciting step forward!
They have addressed some of the editorial comments I provided. However, given the short time period of evaluation and/or small replication, ignoring the behavioral response of the bugs to the aggregation pheromone reduces the confidence of the data. The spillover zone for this lure (at the 1x rate, not the 4x rate which is tested) is already quite large and can cause trap interference. While increased catch in the novel trap design is likely there, the experimental design should incorporate both the objectives of the study and what is already known about the insects behavior. Ignoring the trap spacing that is exclusively used for other studies reduces the repeatability of this dataset. If the study was conducted over a longer period of time or with a year of trap data it would be very interesting to see!
I encourage the authors to continue this research and to conduct a season-long evaluation of the novel trap design, with the appropriate trap spacing.
Author Response
Review 3
Thank you for considering a revision of the manuscript. There is one critical reviewer who has raised some points, which we have addressed, and the following should be useful to explain the difference in perspectives on the purpose of the work. We are grateful for the positive comments from this reviewer.
The authors have developed a novel trap for H. halys which they report catches 7-fold higher number of adults. Although a significant amount of work has occurred on trap design and lure development, there is still a lot of room for growth in this context. The authors trap design looks quite promising for increasing numbers in traps and is an exciting step forward!
Thank you for this.
They have addressed some of the editorial comments I provided. However, given the short time period of evaluation and/or small replication, ignoring the behavioral response of the bugs to the aggregation pheromone reduces the confidence of the data.
We are not ignoring the behavioral response of the bugs to the aggregation pheromone, we are exploiting it. In fact we have exploited it one step more than previous traps - since not only do we attract flying bugs into the funnel inlet, the captured bugs move to the upwind flat mesh panel and do no go near the funnel to escape
The spillover zone for this lure (at the 1x rate, not the 4x rate which is tested) is already quite large and can cause trap interference.
This is acknowledged but it is argued that this is not material to the application. There are also bugs around in the environment releasing the pheromone. Trap pheromone loading also presumably changes due to the arrival of more living bugs releasing the aggregation
While increased catch in the novel trap design is likely there,
Thank you for this.
…the experimental design should incorporate both the objectives of the study and what is already known about the insects’ behavior.
It did so. We wanted to catch the bugs alive. While “Rescue” traps can do this in small numbers, the bugs escape. We are catching large numbers and we think the design prevents escape due to the insect behavior of wanting to escape upwind. Trap overlap needs to be understood in the context of application. Here it is not an issue, but for IPM it is.
Ignoring the trap spacing that is exclusively used for other studies reduces the repeatability of this dataset.
We demonstrate live catch in large numbers for applications such as SIT that we are working on (if there are large number of bugs present). This is reproducible.
If the study was conducted over a longer period of time or with a year of trap data it would be very interesting to see!
We agree.
I encourage the authors to continue this research and to conduct a season-long evaluation of the novel trap design, with the appropriate trap spacing.
Thank you, we plan to. By publishing our new method we expect others to join in conducting this testing. We have already had requests from colleagues to use the traps. We want to put them out in open access as soon as possible.